# Modeling the Radial Growth of Self-Catalyzed III-V Nanowires

**DOI:** 10.3390/nano12101698

**Published:** 2022-05-16

**Authors:** Vladimir G. Dubrovskii, Egor D. Leshchenko

**Affiliations:** 1Faculty of Physics, St. Petersburg State University, Universitetskaya Emb. 13B, 199034 St. Petersburg, Russia; 2Peter the Great St. Petersburg Polytechnic University, Polytechnicheskaya St. 29, 195251 St. Petersburg, Russia; leshchenko.spb@gmail.com

**Keywords:** self-catalyzed III-V nanowires, molecular beam epitaxy, radial growth, modeling

## Abstract

A new model for the radial growth of self-catalyzed III-V nanowires on different substrates is presented, which describes the nanowire morphological evolution without any free parameters. The model takes into account the re-emission of group III atoms from a mask surface and the shadowing effect in directional deposition techniques such as molecular beam epitaxy. It is shown that radial growth is faster for larger pitches of regular nanowire arrays or lower surface density, and can be suppressed by increasing the V/III flux ratio or decreasing re-emission. The model describes quite well the data on the morphological evolution of Ga-catalyzed GaP and GaAs nanowires on different substrates, where the nanowire length increases linearly and the radius enlarges sub-linearly with time. The obtained analytical expressions and numerical data should be useful for morphological control over different III-V nanowires in a wide range of growth conditions.

## 1. Introduction

III-V semiconductor nanowires (NWs) and heterostructures based on such NWs are widely considered as fundamental building blocks for nanoelectronics and nanophotonics [1,2,3,4]. One of the main advantages of freestanding NWs is their ability to relax stress induced by lattice mismatch in NW heterostructures or on a dissimilar substrate such as Si [5,6,7,8], which enables monolithic integration of III-V photonics with Si electronic platforms. Dislocation-free growth of III-V NWs on Si substrates requires, however, sufficiently small lateral dimension. For the InAs-Si material system with a large lattice mismatch of 11.6%, direct growth of InAs on Si is very difficult [9], while InAs NWs with a diameter smaller than 26 nm contain no misfit dislocations [6]. This is one of the reasons behind the importance of controlling the NW radius during growth.

III-V NWs are often obtained by molecular beam epitaxy (MBE) or chemical beam epitaxy using the so-called self-catalyzed vapor-liquid-solid (VLS) method, where the Au catalyst droplet [10] is replaced by a group III (Ga or In) droplet. Self-catalyzed VLS growth, pioneered in Refs. [11,12], is now widely used to produce III-V NWs and III-V NW heterostructures on Si substrates [13,14,15,16,17,18,19,20]. This method has several advantages over the Au-catalyzed VLS growth, including the avoidance of Au contamination and the possibility to grow NWs in regular arrays of patterned pinholes in SiO_x_ mask on Si(111). Ensembles of self-catalyzed III-V NWs can be grown with surprisingly narrow size distributions in terms of both radii [18,21] and length [19,20,22]. Furthermore, the size and shape of the Ga droplet can easily be changed by the group III and V fluxes, giving additional capabilities for the fine tuning of NW morphology [23] and the crystal phase [24]. A detailed review of theoretical and practical aspects of self-catalyzed III-V NWs can be found, for example, in Refs. [8,25]. In some cases, group III droplets can nucleate on top of III-V NWs grown in the catalyst-free selective area mode [26], so understanding the morphological evolution of NWs with droplets on their top is also important for this type of growth. 

The existing models for the radial growth of self-catalyzed III-V NWs [8,18,21] consider a group III droplet as a non-stationary reservoir of group III atoms, which can either swell or shrink depending on the III/V flux ratio and the diffusion flux of group III adatoms to the droplet. The latter depends on the diffusion length of group III adatoms on the NW sidewalls, which has always been a free parameter of modeling. Here, we notice that if (i) group III atoms cannot desorb from the NW sidewalls and droplet [15,27,28], and (ii) the axial NW growth rate is determined by the group V flux, as commonly assumed and confirmed experimentally [11,19,29,30,31], the radial growth of self-catalyzed III-V NWs can be described without any free parameters. Other assumptions of the model are not essential and can be refined. Importantly, we treat the self-catalyzed VLS growth of an ensemble of NWs influenced by the shadowing effect in the MBE technique and the possible re-emission of group III atoms from an oxide mask. Using the recent theoretical results of Refs. [27,28], we develop the fully self-consistent growth model of self-catalyzed III-V NWs, which describes the experimental data on the morphological evolution of GaP and GaAs NWs on different substrates using growth details and no other free parameters. The model should be useful for understanding and controlling the radial growth of different III-V NWs within a wide range of conditions. 

## 2. Model

The main idea of the model is to use the material balance for obtaining the total current of group III atoms into a NW. This current leads to simultaneous axial and radial NW growths. In self-catalyzed III-V NWs, the axial growth rate is given by group V flux. This essential property restricts the applicability of the model to the VLS III-V NW grown from group III droplets or, more generally, from droplets which are enriched with an element constituting the NW itself. For Au-catalyzed III-V NWs, foreign metal catalyzed III-N NWs or ZnO NWs, and catalyst-free III-V or III-N NWs, a more relevant approach has been developed in Refs. [27,28] without using the linear time dependence of the NW length. Any excessive group III flux should necessarily be used for radial growth. The total group III current is limited by group III flux times the surface area per NW, which is why one needs to consider an ensemble of NWs with a given spacing between them. In the first approximation, we treat cylindrical NWs without tapering. A cylindrical shape is a good model for most self-catalyzed III-V NWs growing under steady-state conditions [15,18,19,23,29] without an abrupt increase in V/III flux ratio. For example, the cylindrical shape of self-catalyzed GaAs [23] and GaP [15] NWs is maintained by step flow starting from their top. We also assume a time-independent contact angle of group III droplets catalyzing the NW growth. The droplet volume must increase with time in the presence of radial growth. This process may occur by increasing either the NW radius or the droplet contact angle [24]. For radially extending self-catalyzed GaAs [23,29,30] and GaP [15,31] NWs, and even for Au-catalyzed GaAs NWs with Ga-rich droplets [32], the contact angle is time-independent. This feature is probably explained by surface energetics which favor a certain stable contact angle (around 130° for zincblende GaAs NWs) and actually regardless of the catalyst material [23,24]. At a fixed contact angle and cylindrical NW shape, the two governing equations for the NW volume (determined by the total group III current) and length (determined by group V flux) allow one to obtain a self-consistent equation for the NW radius which contains MBE growth conditions and no other free parameters.

Consequently, here we consider an ensemble of identical vertical III-V NWs having a cylindrical shape with the length L and radius R, grown by MBE on a rotating substrate. Size inhomogeneity in terms of both radii and lengths is neglected. This approximation is relevant for self-catalyzed III-V NWs whose size distributions can be made surprisingly narrow (sub-Poissonian) [18,19,20,21,22]. The contact angle of group III droplets resting on the NW tops equals β, and is kept constant during growth. The surface area per NW equals $cP^{2}$, where P is the pitch of regular array and c is the shape constant (for example, c=1 for square array and $c=\sqrt{3}/2$ for hexagonal array of patterned pinholes). In the case of MBE grown on unpatterned substrates, the surface density of irregular NWs N is related to the pitch as $N=1/\left(cP^{2}\right)$. We treat the most common case of the self-catalyzed VLS growth under effectively group III rich conditions, in which the NW radius enlarges [15,19,23,30]. In terms of the model parameters, this requires that the stationary NW radius introduced hereinbelow is larger than the initial NW radius determined by the size of the initial group III droplets on a substrate surface. 

In the absence of desorption of group III atoms from the NW sidewalls and droplets, the total balance of group III atoms for the VLS growth of NWs on a reflecting substrate is given by
(1)$$\frac{d}{dt}\left[\frac{\pi R^{2}L}{\Omega_{35}}+\frac{1}{\Omega_{3}}\frac{\pi R^{3}}{3}f\left(\beta \right)\right]=\left[S+\left(1-\frac{S}{cP^{2}}\right)S^{\prime }\right]\frac{v_{3}}{\Omega_{35}},\;\frac{S}{cP^{2}}\leq 1,\;\frac{d}{dt}\left[\frac{\pi R^{2}L}{\Omega_{35}}+\frac{1}{\Omega_{3}}\frac{\pi R^{3}}{3}f\left(\beta \right)\right]=cP^{2}\frac{v_{3}}{\Omega_{35}},\;\frac{S}{cP^{2}}>1.$$
with
(2)$$S=2RLtan\alpha_{3}+\frac{\chi_{3}^{}}{cos\alpha_{3}}\pi R^{2},\;S^{\prime }=2RLtan\alpha_{3}+\frac{\chi_{3}^{\prime }}{cos\alpha_{3}}\pi R^{2}.$$

Here, $\Omega_{35}$ is the elementary volume per III-V pair in solid, $\Omega_{3}$ is the elementary volume per group III atom in liquid, f(β)=(1−cosβ)(2+cosβ)/[(1+cosβ)sinβ] is the geometrical function of the droplet contact angle, $v_{3}=v_{3}^{0}cos\alpha_{3}$ is the 2D equivalent MBE growth rate limited by group III flux (with $v_{3}^{0}$ as the total flux of group III atoms per unit time per unit surface area multiplied by $\Omega_{35}$), and $\alpha_{3}$ is the angle of group III beam with respect to the substrate normal. The geometrical coefficients $\chi_{3}$ and $\chi_{3}^{\prime }$ determine the droplet surface areas intercepted by the direct and reflected group III fluxes, and can be obtained using the approach of Ref. [33] as functions of the two angles $\alpha_{3}$ and β. The term $\pi R^{2}L/\Omega_{35}$ in the left side of Equation (1) gives the total number of group III atoms in the NW, which equals the total number of III-V pairs. The term $\pi R^{3}f\left(\beta \right)/\left(3\Omega_{3}\right)$ equals the total number of group III atoms in the droplet, with neglect of group V element due to its very low concentration in the droplet [8,18,21,29]. The S term in the right side of Equation (1) gives the NW and droplet surface area exposed to the direct group III flux. The $\left(1-S/cP^{2}\right)S^{\prime }$ describes the contribution of the group III flux re-emitted from an oxide mask in the case of specular re-emission and accounts for the shadowing effect on the mask surface [27]. When the NW surface area S reaches $cP^{2}$, the mask surface gets fully shadowed by the NW array. After that, each NW receives the maximum group III current $cP^{2}v_{3}$ per unit time, which is determined by the pitch of the surface density of NWs [15,27]. 

The axial growth rate of self-catalyzed III-V NWs is often proportional to the atomic group V flux $v_{5}^{0}$ [8,29]:(3)$$\frac{dL}{dt}=\chi_{5}v_{5}^{0},$$
where $v_{5}^{0}$ is the total flux of group V atoms and $\chi_{5}$ is the geometrical function of the group V beam angle $\alpha_{5}$ and β [33]. Re-emission and desorption of group V atoms from the droplet surface may be included in $v_{5}^{0}$. The 2D equivalent deposition thickness in the thin film case is given by $H=v_{3}t$. We can therefore re-write Equations (1) and (3) in terms of H as follows:(4)$$\frac{d}{dH}\left[\pi R^{2}L+\frac{\Omega_{35}}{\Omega_{3}}\frac{\pi R^{3}}{3}f\left(\beta \right)\right]=S+(1-\frac{S}{cP^{2}})S^{\prime },\;\frac{S}{cP^{2}}\leq 1,$$
(5)$$\frac{d}{dH}\left[\pi R^{2}L+\frac{\Omega_{35}}{\Omega_{3}}\frac{\pi R^{3}}{3}f\left(\beta \right)\right]=cP^{2},\;\frac{S}{cP^{2}}>1,$$
and
(6)$$\frac{dL}{dH}=\frac{\chi_{5}}{cos\alpha_{3}}\frac{v_{5}}{v_{3}}$$

Here, $v_{5}/v_{3}=v_{5}^{0}/v_{3}^{0}$ is the atomic V/III ratio. The factor $\Omega_{35}/\Omega_{3}$ in these equations corrects the corresponding expressions of Refs. [27,28] for the difference in the elementary volumes in the solid and liquid phases (a value on the order of 2). This does not change the results of Refs. [27,28], where the contribution of the droplet volume into the overall material balance was considered negligible. Integration of Equation (6) with the initial condition L(H=0)=0 gives
(7)$$L\left(H\right)=\frac{\chi_{5}}{cos\alpha_{3}}\frac{v_{5}}{v_{3}}H,$$
showing that the length of self-catalyzed III-V NWs is simply proportional to the deposition thickness or growth time [19,23,29]. This growth law may not be valid in the initial stage [15], in which case our model applies starting from the moment of time where the NW length becomes linear in t or H. Using Equation (6) in Equations (4) and (5), we obtain the self-consistent equation describing the NW radius evolution without any free parameters. 

In the case of MBE growth of III-V NWs on an adsorbing substrate on which group III adatoms may diffuse and form the parasitic islands or quasi-2D layer but cannot scatter from the surface, the contribution from re-emission is absent. A fraction of surface adatoms may diffuse from the substrate to the NW base and subsequently to the NW sidewalls, which gives an additional diffusion-induced contribution to the total group III current [28]. When the substrate surface is rough, as occurs during the VLS growth on unpatterned substrates [34] or at low temperatures where slow surface diffusion leads to the formation of parasitic islands on a mask, this contribution is negligible. Furthermore, scattering from a rough surface may be more complex than specular, and more group III atoms may desorb from the substrate surface without landing on the NW sidewalls or droplets. We therefore consider the simplified case of NW growth on an adsorbing substrate without the diffusion exchange between the substrate surface and NWs by simply putting
(8)$$S^{\prime }=0$$
in Equation (4). Clearly, this corresponds to the minimum possible radial growth of self-catalyzed III-V NWs whose axial growth rare is given by Equation (3) and is not influenced by group III flux. 

## 3. Results and Discussion

In the first stage of self-catalyzed VLS growth at $S/\left(cP^{2}\right)\leq 1$, the analytical solution for the NW radius can be obtained only for $S^{\prime }=0$. The governing equation is reduced to
(9)$$\frac{d}{dH}\left[\pi R^{2}L+\frac{\Omega_{35}}{\Omega_{3}}\frac{\pi R^{3}}{3}f\left(\beta \right)\right]=2RLtan\alpha_{3}+\frac{\chi_{3}}{cos\alpha_{3}}\pi R^{2},\;\frac{S}{cP^{2}}\leq 1.$$

Using the linear L(H) dependence given by Equation (7), the NW radius versus length obeys the equation
(10)$$\frac{dR}{dL}=\frac{a+bR/L}{1+cR/L},$$
with
(11)$$a=\frac{sin\alpha_{3}}{\pi \chi_{5}}\frac{v_{3}}{v_{5}},\;b=\frac{1}{2}\left(\frac{\chi_{3}v_{3}}{\chi_{5}v_{5}}-1\right),c=\frac{\Omega_{35}}{\Omega_{3}}\frac{f\left(\beta \right)}{2}.$$

Integration of Equation (10) with the initial condition $R\left(H=0\right)=R_{0}$, where $R_{0}$ is the initial NW radius determined by the size of the droplet resting on the surface, gives
(12)$$L=\frac{R_{0}}{\sqrt{{\left(\frac{R}{L}\right)}^{2}+\frac{\left(1-b\right)}{c}\frac{R}{L}-\frac{a}{c}}}{\left[\frac{2\frac{R}{L}+\frac{1-b}{c}+\frac{A}{c}}{2\frac{R}{L}+\frac{1-b}{c}-\frac{A}{c}}\right]}^{\frac{b+1}{2A}},\;\frac{S}{cP^{2}}\leq 1,$$
where $A=\sqrt{4ac+{\left(1-b\right)}^{2}}$. 

At large enough L, Equation (12) converges to the linear correlation of the NW radius with its length
(13)$$R\to \left[\sqrt{\frac{a}{c}+{\left(\frac{1-b}{2c}\right)}^{2}}-\frac{\left(1-b\right)}{2c}\right]L.$$

Therefore, self-catalyzed III-V NWs grown in an array with a very large pitch (P→∞) would grow radially all the time. Of course, real NW ensembles have a finite inter-NW distance, however large this distance may be, and hence radial growth is stopped when the group III flux reaches its maximum restricted by the surface area per NW $cP^{2}$. 

In the general case with re-emission, Equation (4) can be resolved only numerically. Using again the linear L(H) dependence given by Equation (7), Equation (4) can be presented in the equivalent form
(14)$$\frac{1}{R_{s}^{2}}\frac{d}{dL}\left[\pi R^{2}L+\frac{\Omega_{35}}{\Omega_{3}}\frac{\pi R^{3}}{3}f\left(\beta \right)\right]=\frac{S}{cP^{2}}+\left(1-\frac{S}{cP^{2}}\right)\frac{S^{\prime }}{cP^{2}},\;\frac{S}{cP^{2}}\leq 1,$$
with
(15)$$R_{s}^{2}=\frac{1}{\pi }\frac{cos\alpha_{3}}{\chi_{5}}\frac{v_{3}}{v_{5}}cP^{2}.$$

The first stage of growth ends at $S_{*}=cP^{2}$. From Equation (2), this corresponds to the shadowing length $L_{*}$ and radius $R_{*}$ related by
(16)$$2R_{*}L_{*}tan\alpha_{3}+\frac{\chi_{3}}{cos\alpha_{3}}\pi R_{*}{}^{2}=cP^{2}.$$

The deposition thickness $H_{*}$ is related to $L_{*}$ according to Equation (7). 

In the second, asymptotic stage of growth at $S/\left(cP^{2}\right)>1$, Equation (5) has a simple solution which can be presented in the form
(17)$$L-L_{*}=\frac{L_{*}\left(R^{2}-R_{*}^{2}\right)+\frac{\Omega_{35}}{\Omega_{3}}\frac{f\left(\beta \right)}{3}\left(R^{3}-R_{*}^{3}\right)}{R_{s}^{2}-R^{2}}.$$

The NW radius tends to the stationary radius $R_{s}$ at L→∞. According to Equation (15), the stationary NW radius is proportional to the pitch P and to the III/V flux ratio $v_{3}/v_{5}$. This asymptotic stage of III-V NW growth is independent of the type of the substrate (reflecting or adsorbing) because the substrate surface gets fully shadowed by the NW array and has no influence on the asymptotic growth stage. 

Figure 1 shows the evolution of the NW radii with lengths in the asymptotic stage for three different pitches of 100 nm, 150 nm, and 200 nm. These curves were obtained from Equation (17) assuming the same shadowing length $L_{*}$ of 1000 nm. Other parameters are fixed at $v_{3}/v_{5}=1$, $\alpha_{3}=\alpha_{5}=$ 30°, c=1, and β= 130°. The NW radii $R_{*}$ corresponding to a fixed $L_{*}$ of 1000 nm were obtained from Equation (16). In these dense NW arrays, the NW radius saturates at relatively short lengths. The stationary NW radius is smaller and reached faster for smaller pitches. Figure 2 shows the NW radii versus their lengths in the VLS growth without re-emission at a large pitch of 1000 nm, which has no effect on the morphological evolution for relatively short NWs (≤1000 nm in length). The NWs evolve from the same initial radius of 10 nm; other parameters are fixed at $\alpha_{3}=\alpha_{5}=$ 30°, c=1, and β= 130°. Numerical solutions of Equation (14) at $S^{\prime }=0$ and analytical solutions given by Equation (12) are identical because analytical solution is exact in this case. The radial growth is faster for higher III/V flux ratios $v_{3}/v_{5}$. Decreasing $v_{3}/v_{5}$ has a very substantial impact on suppression of the radial growth. The NW radius reaches almost 100 nm at $v_{3}/v_{5}=$ 1.5, and only 32 nm at $v_{3}/v_{5}=$ 0.5 for the same length of 1000 nm. This effect pertains to individual NWs growing independently of the neighboring NWs in the absence of the shadowing effect. 

Figure 3 shows the NW radii versus lengths in MBE growth on reflecting (Figure 3a) and adsorbing (Figure 3b) substrates at a fixed $R_{0}=$ 20 nm, $v_{3}/v_{5}=0.31$, $\alpha_{3}=\alpha_{5}=$ 30°, c=1, and three different pitches of 300 nm, 500 nm, and 700 nm. At a given length and pitch, the NWs growing on a reflecting substate are thicker than the ones growing on an adsorbing substrate due to the additional re-emitted flux of group III atoms. At a given pitch, the NW radii converge to the same stationary value in the asymptotic growth stage. The NW radii vary with the pitch even before the full shadowing of a reflecting mask in Figure 3a, because the re-emitted flux of group III atoms is affected by the shadowing effect from the very beginning of growth. Conversely, the radii of NWs growing on an adsorbing substrate (without surface diffusion of group III adatoms from the substrate surface to NWs) are exactly identical before the full shadowing of the substrate in Figure 3b. This is explained by the absence of material exchange between the substrate surface and NWs. The NW radii become different only when the shadowing effect starts to affect the bottom part of NWs which no longer receive any flux. In both cases, the NWs grow considerably thinner for smaller pitches. This suppression of the radial growth is entirely due to the shadowing effect in the directional MBE technique. 

Based on these theoretical results, one can consider three ways to reduce radial NW growth. First, the V/III flux ratio can be increased to partly consume the droplet and decrease the flux impinging the NW sidewalls. This method works regardless of the pitch and even for isolated NW, as noticed previously in Refs. [18,21,23]. Second, radial growth can be suppressed by decreasing the pitch or increasing NW number density, as a result of the shadowing effect [15,27,28]. Third, NWs grow thinner on adsorbing substrates due to the absence of any re-emitted group III flux, although at a cost of parasitic growth between the NWs. This conclusion is supported in the next section. 

## 4. Theory and Experiment

Ga-catalyzed GaP NWs of Ref. [15] were grown by MBE at 600 °C in a regular hexagonal array of lithographically patterned pinholes in SiO_2_ mask on Si(111) substrate ($cP^{2}=$ 216,500 nm^2^), with Ga pre-deposition. The growth was performed at $\alpha_{3}=\alpha_{5}=$ 32°, $v_{3}=$ 0.135 nm/s, and $v_{3}/v_{5}=0.37$ for 60 min. 122 GaAsP markers were introduced at fixed time intervals to monitor ex-situ the axial growth, radial extension, and Ga droplet swelling of individual NWs. The droplet contact angle stayed nearly constant at β= 135°, corresponding to the region of zincblende GaP NWs where the droplet enlarges by increasing its base radius. An almost untapered NW geometry was kept by step flow radial growth starting from the NW top. The NW axial growth rate increased linearly with the deposition thickness H from 0.3 nm/s at the beginning to 1 nm/s at H= 105.3 nm, and then stayed constant at 1 nm/s until the end of growth at H= 486 nm. Therefore, our model applies only to the interval of H from 105.3 nm to 486 nm, corresponding to the interval of NW length from ~500 nm to ~3300 nm. 

Analysis of these data led to the conclusion of specular re-emission of Ga atoms from SiO_2_ mask being the mechanism of material exchange between the substrate and NWs [15]. This conclusion was also supported by the growth modeling of Ref. [27]. Figure 4a shows the comparison of the model with the data of Ref. [15]. The theoretical curve was obtained from Equation (14) with the growth parameters listed above. Figure 4b shows the comparison of our solution with the quadratic approximation used in Ref. [15] for the average NW radii. Without any free parameters, the correspondence with the data in both figures is very good. The radius–length dependence is almost linear for short lengths, converging to a saturating curve for larger lengths due to the shadowing effect.

Ga-catalyzed GaAs NWs of Ref. [19] were grown by MBE at 640 °C. A special lithography-free procedure was used for the preparation of pinholes in the oxide layer on Si(111), and the NW growth was performed without pre-deposition of Ga. This resulted in random positioning of NWs, with an average surface density $N=1/\left(cP^{2}\right)$ = 10^8^ cm^−2^. The Ga beam angle was 35°, and the 2D equivalent growth rate $v_{3}$ was 0.1 nm/s. Six NW samples were grown using different growth durations of 3 min, 6 min, 20 min, 40 min, 50 min, and 80 min, and the average length and radii were obtained by measuring more than 30 NWs per sample. Best linear fit to the average length versus time yielded a time-independent axial NW growth rate of 1.05 nm/min. The average contact angle of Ga droplets was estimated at 133°, corresponding to the zincblende crystal phase. The III/V flux ratio $v_{3}/v_{5}=0.214$ was obtained from the NW axial growth rate at the known $v_{3}$. All NWs had uniform radii from base to top. Figure 5a shows the average NW radius versus average length, and the curves obtained from Equation (14) with Ga re-emission are included. Figure 5b shows the time dependences of the mean NW length and radius. The evolution of the mean NW length with time is almost perfectly linear. Furthermore, the size inhomogeneities in terms of both radii and lengths are very small. These two features support the model assumptions.

Ga-catalyzed GaAs NWs of Ref. [30] were grown by MBE on unpatterned Si-treated GaAs(111)B substrates at 640 °C and a V/III flux ratio of 5. The 2D equivalent growth rate $v_{3}$ was 0.278 nm/s at $\alpha_{3}=$ 30°. Substantial parasitic growth occurred on the substrate surface between the NWs. The NW length scaled linearly with time. The droplet contact angle is estimated at 135° from the linear fit of the NW length versus time. The NW surface density, estimated from the images of Ref. [30], corresponds to $cP^{2}=$ 180,000 nm^2^. The average lengths and radii of the NWs were measured on the samples grown for different times. Figure 6 shows a comparison of the experimental data with the models given by Equations (12) and (14) without and with Ga re-emission from the substrate surface, respectively. It is shown that the curve without Ga re-emission has better correspondence with the data, apart from one experimental point which does not follow the general trend and probably corresponds to a sample grown under slightly different conditions. The curve with Ga re-emission largely overestimates the NW radii in all cases. This confirms our theoretical conclusion of the NW radius being smaller in the absence of re-emitted Ga flux. 

Overall, the radii of self-catalyzed III-V NWs in Figure 4–6 show a similar sublinear increase with the NW length for long enough growth times. For the morphology of GaP and GaAs NW arrays grown in SiO_x_ pinholes on Si(111) substrates, the data are well described using the model with specular re-emission of Ga atoms from the mask surface. Therefore, this mechanism of material exchange between the mask surface and NWs should be considered dominant, as discussed earlier in Ref. [15]. The radial growth of GaAs NWs on unpatterned GaAs(111)B substrates is better described using the model without Ga re-emission. In this case, Ga adatoms are quickly trapped by the parasitic layer and cannot diffuse to the NWs, similarly to Ref. [34]. When the NW length scales linearly with time, the NW radius may increase linearly or even super-linearly in the initial growth stage, converging to sub-linear growth saturating to a constant determined by the pitch of the NW array. The radial growth rate is lower for smaller pitches, a larger V/III flux ratio, and in the absence of Ga re-emission from the substrate surface. 

## 5. Conclusions 

In summary, a self-consistent growth model was developed for the morphological evolution of the ensembles of self-catalyzed III-V NWs, which includes the possible re-emission of group III atoms from the mask and the shadowing effect in the directional MBE technique. The model contains no free parameters and describes the radial growth of self-catalyzed III-V NWs on different substrates. The model describes quite well the data on the growth kinetics of self-catalyzed GaP and GaAs NWs on Si(111) and GaAs(111)b substrates. Specular re-emission of Ga atoms from silica masks has been identified as the main mechanism of material exchange between the substrate surface and NWs grown on Si(111) substrates. GaAs NWs grown on unpatterned GaAs(111)B substrates are better described using the model without any material exchange between the substrate and NWs. The model applies to cylindrical NWs with a uniform radius from base to top. We now plan to develop this approach and consider more delicate growth scenarios where the droplet volume can enlarge by increasing its contact angle or the NW can taper depending on the material fluxes. 

## Figures and Tables

**Figure 1 nanomaterials-12-01698-f001:**
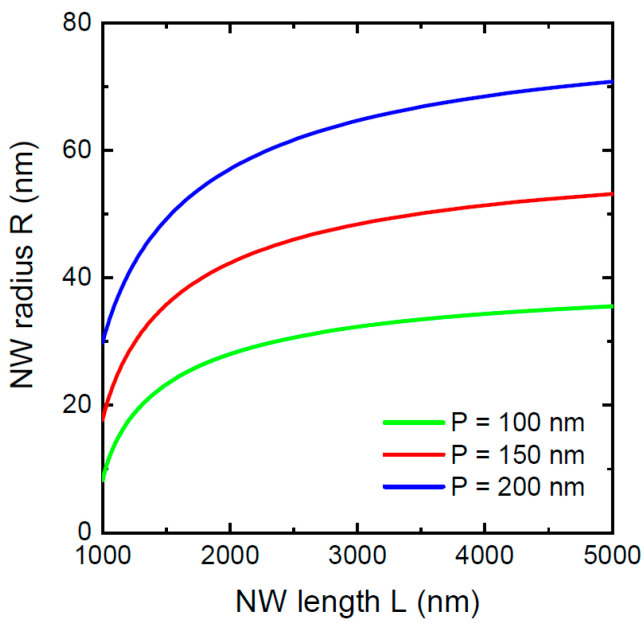
Evolution of the NW radius with length in the asymptotic growth stage, obtained from Equation (17) at a fixed $L_{*}=$ 1000 nm (corresponding to different $R_{*}$), $v_{3}/v_{5}=1$, $\alpha_{3}=\alpha_{5}=$ 30°, c=1, and β= 130°, and three different pitches P  shown in the legend.

**Figure 2 nanomaterials-12-01698-f002:**
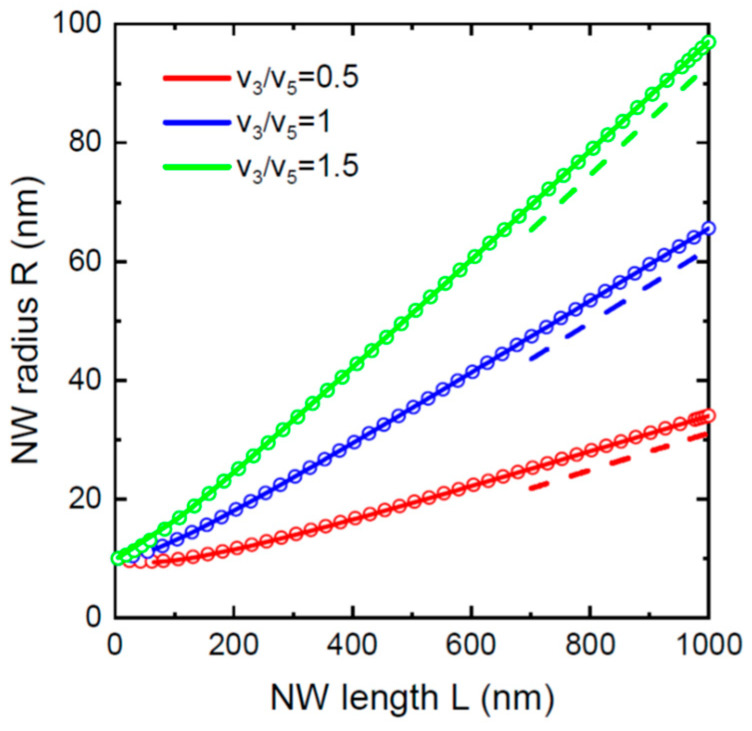
Evolution of the NW radius with length at a large pitch P of 1000 nm, fixed $R_{0}=$ 10 nm, $\alpha_{3}=\alpha_{5}=$ 30°, c=1, and β= 130°, for three different III/V flux ratios $\;v_{3}/v_{5}$ shown in the legend. Solid lines correspond to analytical solution given by Equation (12), symbols show the numerical solution of Equation (14) at $S^{\prime }=0$, and dashed lines are the asymptotes given by Equation (13).

**Figure 3 nanomaterials-12-01698-f003:**
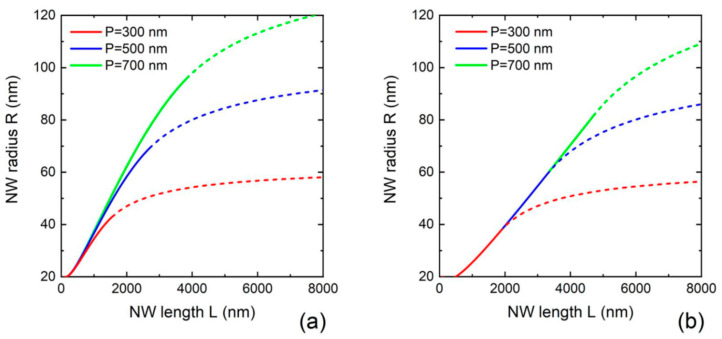
Evolution of the NW radius with length in MBE growth on (**a**) reflecting and (**b**) adsorbing substrates for three different pitches P shown in the legends. The curves are obtained from Equation (14) in (**a**) and Equation (12) in (**b**) at the fixed $R_{0}=$ 20 nm, $v_{3}/v_{5}=0.31$, $\alpha_{3}=\alpha_{5}=$ 30°, c=1, and β= 135°. The solid and dashed lines show the morphological evolution before and after the full shadowing of the substrate surface, respectively.

**Figure 4 nanomaterials-12-01698-f004:**
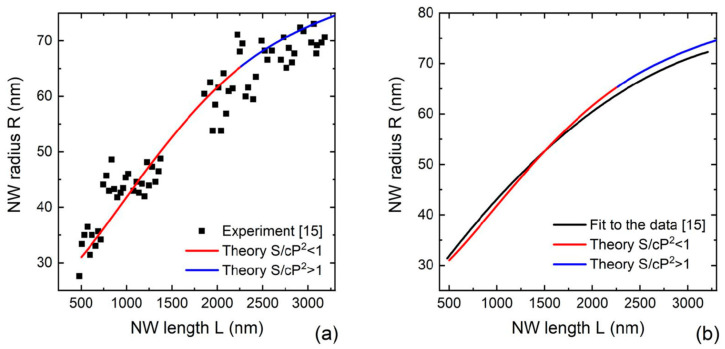
(**a**) Comparison of the model solution with the data of Ref. [15] on the morphological evolution of Ga-catalyzed GaP NWs grown in regular array on SiO_2_/Si(111) surface. (**b**) Same as (**a**) for the average NW radius, fitted by quadratic time dependence in Ref. [15]. The red and blue lines correspond to the solutions before and after the full shadowing of the silica mask, respectively.

**Figure 5 nanomaterials-12-01698-f005:**
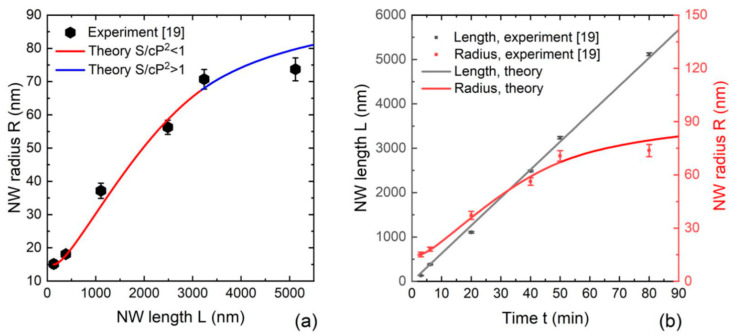
(**a**) Average radius of Ga-catalyzed GaAs NWs versus length [19] (symbols) compared to the solution given by Equation (14). (**b**) Average NW length and radius versus growth time. The red and blue sections in (**a**) correspond to the solutions before and after the full shadowing of the silica mask, respectively.

**Figure 6 nanomaterials-12-01698-f006:**
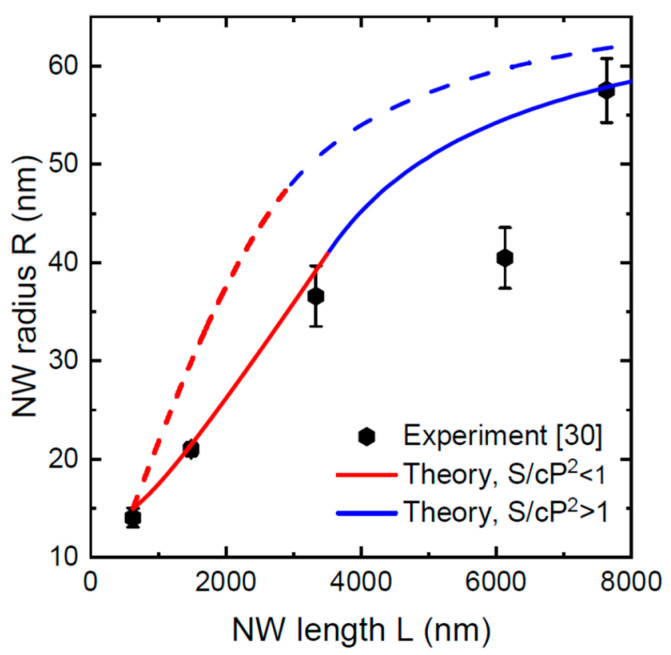
Comparison of the model solutions without (solid line) and with (dashed line) Ga re-emission with the data on the morphological evolution of Ga-catalyzed GaAs NWs grown by MBE on unpatterned GaAs(111)B substrates [30]. The red and blue sections correspond to the solutions before and after the full shadowing of the silica mask, respectively.

## Data Availability

Not applicable.

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
