# Peer review of "Modeling the Radial Growth of Self-Catalyzed III-V Nanowires"

_nanomaterials, 2022, doi:10.3390/nano12101698_

Round 1
Reviewer 1 Report
This manuscript, the authors study the models for the radial growth of self-
catalyzed III-V NWs combining the theory and experiment investigation. It was
found that the reported growth model could well present the fully self-
the consistent growth model of self-catalyzed III-V NWs, which well agrees with
the previous experiment investigation. In a summary, the results are
interesting, could well understand the growth details. Therefore, the reviewer
considers that the manuscript can be accepted after a revision.
1. Controllable growth of materials with high quality is well required. The
current model is based on the growth of III-V nanowires. The question is
does the model also consistent with other materials. For example,
recently it was found that salt could effectively catalyze the growth of van
der Waals materials. ACS AMI, 2021, doi.org/10.1021/acsami.1c19906
2. Normally the growth of the nanowire also affected by its growth direction.
Does the model consider the used substrate and the growth direction,
such as in-plane direction along the substrate or out-of-plane direction
vertically with the substrate?
3. Does this model also consist of the GaN nanowire or ZnO nanowire?
Author Response
Please see the response letter attached

Reviewer 2 Report
This manuscript mainly demonstrates a new model for the radial growth of self-catalyzed III-V nanowires on different substrates. This model mainly focuses on the possible re-emission of group â…¢ atoms from the mask and the shadowing effect in the directional MBE technique. The data of Self-catalyzed GaP and GaAs NWs on Si substrates are obtained. However, I don’t think this manuscript is suitable for publication in nanomaterials. My comments are as follows:
- The English of this manuscript needs to be improved and the writing format needs to be carefully checked. For example, on page 9, line 280, there are two “and”. Too many clauses in the manuscript make it difficult to understand.
- The derivation process and idea of the formula are not very clear and convincing. Most of the content was to introduce the meaning of parameters.
- The laws summarized in Figures 1, 2, and 3 are not supported by a particularly large amount of data, which makes reviewers feel that there are conclusions before data.
- The amount of data referred to in model fitting is small, and only two references are involved in total. Such fitting results are not very convincing.
Author Response
Please see the response letter attached.

Reviewer 3 Report
The paper presents a theoretical model for the radial growth of self-catalyzed III-V nanowires on different substrates, i.e., surfaces reflecting or adsorbing group III atoms flux. The overall quality is very good, and the paper is written in good English. The Authors consistently explained the model, and the general agreement with experimental data is good. A few minor issues must be addressed, more in detail:
- One of the main assumptions for the proposed model is the constancy of the droplet contact angle. As also stated in 10.1088/1361-6528/abb106, by one of the Authors, this is not always true. More references to experimental studies are necessary to support the assumption further.
- 4 and 5: the curves do not fit the experimental points, but they compare the presented models with the experimental data using growth details and no other free parameters. The curves describe (better than “fitting”) very well the experimental data, and the model reproduces the observed growth. The same comment holds for the use of “fits” in line 13.
The number of self-references is 13 on 31, about 42% of all paper citations. Even if we consider the Authors established experience in the research field, it would be helpful to reduce this percentage by adding other critical experimental references, as previously suggested.
Author Response

(The authors gave the same response as above.)

Round 2
Reviewer 2 Report
The problems have been solved.